# Menopause as a Critical Turning Point in Lipedema: The Estrogen Receptor Imbalance, Intracrine Estrogen, and Adipose Tissue Dysfunction Model

**DOI:** 10.3390/ijms26157074

**Published:** 2025-07-23

**Authors:** Diogo Pinto da Costa Viana, Lucas Caseri Câmara, Robinson Borges Palau

**Affiliations:** 1Brazilian Society of Endocrinology and Metabolism in Sports and Exercise, Florianópolis 88070-800, Brazil; 2Brazilian Society of Obesity Medicine, Florianópolis 88070-800, Brazil; 3Department of Specialization in Clinical Anabolism College of Governance, Engineering, and Education of São Paulo-FGE-SP, São Paulo 18705-770, Brazil

**Keywords:** lipedema, menopause, estrogen receptors, ERα and Erβ, intracrine estrogen, adipose tissue dysfunction, aromatase, 17β-hydroxysteroid dehydrogenase

## Abstract

Lipedema is a chronic, estrogen-sensitive adipose tissue disorder characterized by disproportionate subcutaneous fat accumulation, fibrosis, inflammation, and resistance to fat mobilization. Despite its high prevalence, lipedema remains poorly understood and frequently misdiagnosed. This narrative review proposes a novel pathophysiological model in which menopause acts as a critical turning point in the progression of lipedema, driven by estrogen receptor imbalance (ERβ predominance over ERα), intracrine estrogen excess, and adipose tissue dysfunction. We demonstrate how menopause amplifies adipose tissue dysfunction by suppressing ERα signaling; enhancing ERβ activity; and disrupting mitochondrial function, insulin sensitivity, and lipid oxidation. Concurrently, the upregulation of aromatase and 17β-HSD1, combined with the suppression of 17β-HSD2, sustains localized estradiol excess, perpetuating inflammation, fibrosis, and immune dysregulation. The molecular signature observed in lipedema closely mirrors that of other estrogen-driven gynecological disorders, such as endometriosis, adenomyosis, and uterine fibroids. Understanding these molecular mechanisms highlights the pivotal role of menopause as a catalyst for disease progression and provides a rationale for targeted therapeutic strategies, including hormonal modulation and metabolic interventions. This review reframes lipedema as an estrogen receptor-driven gynecological disorder, offering a new perspective to improve clinical recognition, diagnosis, and management of this neglected condition.

## 1. Introduction

Lipedema is an inflammatory disorder of subcutaneous adipose tissue, characterized by a disproportionate, symmetrical, and painful accumulation of fat that is resistant to caloric restriction [1]. This condition remains unresponsive to conventional weight loss methods and is often triggered or exacerbated by hormonal changes. In contrast, obesity represents a generalized excess of body fat that typically improves with dietary and lifestyle interventions. Differentiating between these two conditions is essential for accurate diagnosis and effective treatment. Lipedema is hypothesized to follow a progressive clinical course, although this has not been definitively confirmed and remains based primarily on clinical observations [2]. In Europe, prevalence estimates range widely—from 0.06% to 39%—due to inconsistent diagnostic criteria, varying data collection methods, and frequent underreporting [3]. Despite the lack of standardized global data, the condition is estimated to affect approximately 10% of women, contributing to the high rate of underdiagnosis [2]. In Brazil, prevalence among women has been reported at around 12.3%, reinforcing the need for greater clinical awareness and standardized diagnostic protocols [3].

Lipedema has a strong hormonal component, with evidence suggesting an imbalance in the distribution of estrogen receptors within adipose tissue, leading to adipocyte hypertrophy, inflammation, and fibrosis [4]. The condition tends to manifest or worsen during periods of significant hormonal fluctuations in a woman’s life, such as menarche, pregnancy, and menopause [5,6]. Menopause represents a critical window for the onset or exacerbation of cardiometabolic alterations, including increased insulin resistance, dyslipidemia, and low-grade systemic inflammation. These changes are accompanied by a progressive redistribution of body fat, particularly an increase in visceral adipose tissue, even in the absence of significant weight gain [7] (Figure 1).

As a result, the prevalence of metabolic syndrome rises substantially during the perimenopausal period, potentially interacting synergistically with the inflammatory microenvironment of lipedema-affected adipose tissue. This interplay promotes endothelial dysfunction, fluid retention, pain, and further resistance to fat mobilization [8]. Recent transcriptomic analyses have identified specific microRNAs dysregulated in lipedema adipose tissue, highlighting a distinct epigenetic profile associated with endocrine resistance and impaired adipocyte homeostasis [9].

Fluctuations in systemic estradiol levels during the menopausal transition also profoundly alter adipose tissue metabolism by shifting estrogen receptor expression. This shift favors a predominance of estrogen receptor beta (ERβ) over alpha (ERα), thereby amplifying local inflammation and aggravating lipedema [4,5,6,7,8].

Longitudinal studies indicate that the menopausal transition is associated with both structural and functional vascular changes, including increased carotid intima-media thickness, arterial stiffness, and endothelial dysfunction [10]. Combined with the loss of estradiol’s vasodilatory and anti-inflammatory effects, these alterations may further impair the already compromised microvascular and lymphatic systems in patients with lipedema, worsening venous congestion, edema, and the persistence of chronic pain.

Estradiol modulates adipose tissue metabolism by promoting mitochondrial integrity, activating PI3K/Akt signaling, and attenuating fibrotic responses—mechanisms severely compromised in the menopausal transition [11].

This review critically examines how hormones, especially during menopause, influence lipedema progression. It highlights estrogen imbalance, local hormone production, progesterone resistance, and hormonal deficiency, along with treatment implications for menopausal women.

## 2. Results

This review is based on an integrative pathophysiological model derived from observational data, molecular biology, and extrapolations from related estrogen-dependent disorders. Despite the biological plausibility and strong mechanistic rationale, the lack of specific randomized clinical trials in lipedema populations represents a limitation. Future studies should aim to validate these mechanisms and the proposed therapeutic interventions.

## 3. Discussion

### 3.1. The Role of Estradiol and Its Receptors

The effect of estradiol on tissues is not absolute; rather, it depends on the functional balance between its two main nuclear receptors, ERα and ERβ, which often exert opposing yet complementary actions. ERα exhibits a predominantly anabolic, proliferative, and homeostatic profile, promoting healthy expansion of subcutaneous adipose tissue, enhanced insulin sensitivity, extracellular matrix integrity, and cardiovascular protection. In contrast, ERβ functions as a negative modulator, exerting antiproliferative and antiadipogenic effects, and serves as a physiological brake on ERα activity. During menopause and in conditions like lipedema, there is a shift in the ERα/ERβ expression profile—marked by ERα downregulation and ERβ upregulation. This imbalance disrupts the anabolic, insulin-sensitizing, and anti-inflammatory actions of estradiol, contributing to adipocyte dysfunction, fibrosis, and chronic inflammation [8,12,13]. The receptor imbalance also contributes to a loss of cardiovascular protection, metabolic deterioration, and preferential accumulation of visceral fat. The interdependence of these receptors is evident in their competition for the same estrogen response elements (EREs) in DNA and for binding to the same estradiol molecule, dynamically modulating the hormone’s biological effects.

Another key aspect is the capacity of subcutaneous adipose tissue to locally synthesize estrogens via steroidogenic enzymes such as aromatase (CYP19A1) and 17β-HSD, supporting a sustained autocrine/paracrine cycle of estrogenic activation. This local estrogen production may intensify ERβ signaling while diminishing ERα activity, thereby promoting the progressive growth of affected adipose depots. The convergence of a locally imbalanced receptor profile, enhanced intratissue estrogen synthesis, and dysregulated coregulator expression establishes a permissive environment for the progression of lipedema.

It is important to note that much of the mechanistic insight discussed in this section originates from research on obesity models. Although these studies provide valuable frameworks, lipedema adipocytes demonstrate unique metabolic and hormonal characteristics—including altered insulin sensitivity, resistance to caloric restriction, and distinct inflammatory responses—that may limit the direct applicability of obesity-based findings to lipedema pathophysiology [4,8,12]. Therefore, while biologically plausible, these mechanisms should be validated in lipedema-specific populations.

### 3.2. Intracrine Production of Estradiol in Adipose Tissue

Adipose tissue functions not only as a hormonal target but also as an active site of estrogen synthesis. Local estrogen production and intracrine signaling through ERα and ERβ directly influence lipogenesis, lipolysis, and tissue integrity in both pre- and postmenopausal women, supporting the concept of an autonomous and functional hormonal microenvironment [14]. In lipedema, enzymatic dysregulation occurs within the affected adipose tissue: the expression of 17β-hydroxysteroid dehydrogenase type 1—enzymes responsible for converting estrone into active estradiol—is increased twofold in lipedema adipocytes [15,16], alongside heightened aromatase (CYP19A1) activity. This enzymatic asymmetry establishes a locally hyperestrogenic microenvironment within lipedematous tissue, sustaining a cycle of pathological adipogenesis even in the context of low systemic estradiol levels [4,16,17].

During menopause, there is an android redistribution of fat driven by decreased estradiol levels. Simultaneously, lipedematous tissue preserves local hyperestrogenism through the overexpression of aromatase and 17β-HSD1, preferentially activating ERβ, which promotes dysregulated adipogenesis and fibrosis [4].

The dysregulation of nuclear coregulators may, therefore, represent a central axis in the pathophysiology of lipedema [5].

Under physiological conditions, a functional balance exists between ERα and ERβ receptors. However, in women with lipedema, this equilibrium is lost. Histological studies reveal that in affected regions, ERα expression is diminished while ERβ is relatively upregulated, resulting in altered local sensitivity to estrogenic action and promoting a phenotype resistant to fat mobilization [4,16,17].

Estrogen signaling in adipose tissue plays a critical role in regulating female fat distribution, with ERα enhancing lipogenesis and triglyceride storage particularly in the gluteofemoral regions. This regulation occurs via increased free fatty acid uptake through lipoprotein lipase (LPL) activation and suppression of lipolysis through upregulation of α2A-adrenergic receptors, along with enhanced angiogenesis via vascular endothelial growth factor (VEGF). ERβ dominance, commonly observed in lipedema, inhibits mitochondrial biogenesis and reduces the tissue’s oxidative capacity [4].

Menopause directly contributes to this dysregulation. Several studies have shown that during the climacteric period, ERα expression is downregulated while ERβ is compensatorily upregulated in adipose tissues, further exacerbating estrogen receptor imbalance in women predisposed to lipedema [8,18]. To summarize the key molecular changes associated with hormonal imbalance in lipedema, Table 1 presents an overview of the main pathophysiological alterations, their corresponding enzymes and receptors, and supporting references from the literature.

### 3.3. Progesterone Resistance

Progesterone plays a fundamental protective role in adipose tissue homeostasis by regulating adipocyte differentiation, modulating inflammation, and inhibiting steroidogenic enzymes [18]. However, in lipedema, there is consistent evidence of tissue-level progesterone resistance, particularly in the affected regions.

This resistance is marked by reduced expression and activity of 17β-HSD2, the enzyme responsible for converting active estradiol into inactive estrone. The deficiency of this enzyme allows estradiol to remain biologically active for an extended period, thereby promoting continuous estrogenic signaling via ERβ and exacerbating adipogenesis and chronic inflammation.

Lipedema shares key pathophysiological mechanisms with several estrogen-dependent gynecological conditions, including endometriosis, adenomyosis, and uterine fibroids [15]. During the menopausal transition, the onset or worsening of these disorders is frequently observed due to dysregulation of the hypothalamic–pituitary–ovarian axis—an endocrine instability that also contributes to the aggravation of lipedema. However, unlike endometriosis and uterine fibroids, which typically regress after menopause due to systemic estrogen depletion, lipedema progression appears to persist or even intensify, driven by local estrogen production through intracrine mechanisms and an imbalance favoring ERβ over ERα signaling. These mechanisms involve increased activity of 17β-HSD1 and aromatase, which convert estrone (E_1_) into estradiol (E_2_), along with reduced 17β-HSD2 activity due to progesterone resistance. Although evidence from lipedema-specific studies remains limited, these mechanisms are supported by findings in related estrogen-sensitive disorders such as endometriosis and fibroids, which may offer pathophysiological parallels. This disruption impairs the inactivation of E_2_ back into E_1_, leading to local estradiol accumulation within adipose tissue. The resulting estradiol trapping enhances ERβ-mediated signaling, promoting inflammation, fibrosis, and pathological expansion of adipose tissue—core features of lipedema [19,25]. This mechanism establishes a self-perpetuating cycle of disease progression. Progesterone fails to perform its stabilizing function, and intracrine estradiol maintains the adipose tissue in an activated, proinflammatory state that is resistant to regression—a pattern characteristic of lipedema, particularly after menopause. Elevated estradiol activity in progesterone-resistant adipose tissue perpetuates a hormonal environment conducive to the advancement of the disease [4,15].

### 3.4. How Menopause Affects Lipedema

The menopausal transition is characterized by a progressive decline in estradiol levels and a shift in the balance of estrogen receptors, with decreased ERα activity and a relative increase in ERβ. This imbalance disrupts the homeostasis of female adipose tissue and initiates metabolic, inflammatory, and structural changes affecting both subcutaneous and visceral fat depots.

From a metabolic perspective, estrogen deficiency leads to a redistribution of body fat from a gynoid to an android pattern, with increased visceral fat accumulation—even in the absence of significant weight gain. This phenomenon is associated with elevated lipoprotein lipase (LPL) activity in visceral adipose tissue, tissue hypoxia (↑HIF-1α), heightened inflammation (↑IL-6, IL-18), and impaired lipogenesis [8,26].

Menopausal estrogen decline induces a state of low-grade chronic inflammation characterized by activation of the NF-κB and JNK pathways, increased infiltration of pro-inflammatory M1 macrophages, and elevated levels of cytokines such as TNF-α, IL-6, and IL-1β. In lipedema, this inflammatory profile is further amplified by a phenotypic shift from anti-inflammatory M2 (CD163^+^) macrophages to M1 dominance, which accelerates fibrosis, impairs lymphatic flow, and fosters progressive adipose tissue dysfunction and metabolic deterioration [15,23,24,27].

In the context of lipedema, this new hormonal and metabolic configuration acts as a catalyst for disease progression. The imbalance between ERα (↓) and ERβ (↑)—described in both menopause and lipedema—amplifies anti-lipolytic, pro-inflammatory, and pro-fibrotic effects in subcutaneous adipose tissue [28,29]. This scenario promotes PPARγ activation, increased glucose uptake (via GLUT4), and enhanced free fatty acid uptake (via LPL), in association with suppressed lipolysis (↑αAR/↓βAR) and mitochondrial dysfunction.

Another key mechanism is the intensification of intracrine estradiol production in the affected adipose tissue, driven by the overexpression of aromatase and 17β-HSD1. In the absence of systemic hormonal regulation typical of the reproductive years, this local pathway perpetuates adipogenesis, inflammation, and fibrosis. The coexisting progesterone resistance further exacerbates this condition by reducing the activity of 17β-HSD2, the enzyme responsible for converting active estradiol into inactive estrone, thereby amplifying local estrogenic signaling [4,15,16].

Clinically, both the worsening of pre-existing conditions and the onset of late-onset lipedema after menopause are observed, often presenting in more severe forms that are resistant to conventional treatments [30]. This pattern is directly related to the new hormonal environment of the climacteric, characterized by systemic estradiol deficiency, ERβ dominance, progesterone resistance, and low-grade chronic inflammation. Recognizing this pattern is essential for accurate diagnosis and early intervention, particularly in women during perimenopause or early menopause.

Even in the absence of weight gain, menopause induces a preferential redistribution of fat to the visceral compartment, with a relative reduction in fat in the lower limbs. In lipedema, where subcutaneous tissue already presents inflammation, impaired lipid mobilization, and metabolic dysfunction, the loss of estrogenic regulation exacerbates these alterations, intensifying fat accumulation in already compromised regions [13,31].

Therefore, in the context of lipedema, menopause not only exacerbates pre-existing mechanisms—such as inflammation, fibrosis, lymphatic dysfunction, and resistance to lipid mobilization—but also acts as a trigger for the emergence of more severe and treatment-resistant forms of the disease. These pathophysiological changes and their clinical implications are illustrated in Figure 2.

### 3.5. Therapeutic Implications

The understanding of menopause as a catalyst for the progression of lipedema profoundly redefines the therapeutic approach to this condition. The identification of specific hormonal and metabolic targets, particularly in the context of estrogen deficiency during the climacteric, enables more personalized, preventive, and potentially more effective strategies.

Hormone replacement therapy, especially with transdermal bioidentical estradiol, represents a cornerstone intervention in protecting adipose tissue from the deleterious changes induced by menopause. Robust evidence indicates that estradiol, through predominant activation of the ERα receptor, plays a central role in maintaining metabolic homeostasis, modulating inflammation, and preventing tissue fibrosis [8,32].

The timing of HRT initiation is a critical determinant of its therapeutic efficacy. Initiating estrogen replacement early—during the menopausal transition or within the first few years post-menopause—is essential to prevent dysregulation of the estrogen receptor balance, characterized by a decline in ERα and a relative increase in ERβ. This imbalance plays a central role in the pathogenesis of inflammation, fibrosis, and lymphatic dysfunction observed in lipedema [10,31]. This concept, consistent with the timing hypothesis of HRT already established in cardiovascular and neurocognitive health, should likewise be applied to the pathophysiology of adipose tissue.

Beyond the therapeutic window, the route of administration and the site of estradiol application are important clinical considerations in the management of lipedema. Given the heightened intracrine estradiol production within the affected adipose tissue—sustained by increased expression of aromatase and 17β-HSD1 and diminished activity of 17β-HSD2—the preferred approach is the use of transdermal formulations (patches or gel) applied to areas not affected by lipedema, such as the back, scapular region, or inner arms (if disease-free). This strategy minimizes local stimulation of dysregulated steroidogenesis, thereby contributing to better control of inflammation and fibrosis.

The choice between progestins and natural progesterone should be guided by the stage of the climacteric and the presence of coexisting gynecological conditions. During the menopausal transition (STRAW stages −2 and −1), progestins in combination with estradiol may offer advantages due to fluctuating endogenous estrogen levels. Progestins can attenuate the amplitude of pulsatile LH and FSH secretion; reduce the risk of HRT-associated bleeding; and demonstrate greater therapeutic efficacy in women with gynecological disorders such as adenomyosis, endometriosis, and uterine fibroids. Among the progestins with potential efficacy in lipedema are two notable agents: drospirenone and gestrinone [26,33].

Drospirenone is a fourth-generation progestin derived from spironolactone, exhibiting antiandrogenic and antimineralocorticoid activity, along with modulatory effects on inflammation and adipose tissue metabolism. Its primary benefit in lipedema lies in its ability to counteract progesterone resistance—a key pathogenic feature of the disease—while also exerting direct anti-inflammatory effects via activation of progesterone receptors, particularly PRβ. This activation results in reduced expression of pro-inflammatory cytokines such as TNF-α and IL-1β and enhanced production of anti-inflammatory mediators such as IL-10 [15,34].

In addition, drospirenone exerts a significant anti-adipogenic effect by inhibiting the differentiation of preadipocytes and triglyceride accumulation, primarily through antagonism of mineralocorticoid receptors [35]. This mechanism directly influences the pathological microenvironment of lipedema, contributing to reduced fluid retention, chronic inflammation, and fibrosis—particularly in more advanced stages of the disease. Evidence from studies in postmenopausal women suggests that the combination of drospirenone with estradiol improves body fat distribution, reduces central fat mass, and enhances the adipokine profile—effects that are highly relevant for the metabolic and tissue management of lipedema [15,35,36].

Gestrinone is a synthetic progestogen with potent modulatory action on intracrine steroidogenesis, widely used for decades in the treatment of estrogen-dependent gynecological disorders such as endometriosis, adenomyosis, and uterine fibroids. Its therapeutic relevance in lipedema derives from the shared pathophysiological mechanisms among these conditions, particularly in the context of progesterone resistance, aromatase overexpression, and enzymatic imbalance—marked by increased 17β-HSD1 and reduced 17β-HSD2. Although increased aromatase and 17β-HSD1 activity in lipedema tissue has been described in some histological studies, the direct causal impact of these enzymatic patterns on disease progression remains to be fully validated in human clinical models.

Gestrinone directly inhibits aromatase and 17β-HSD1 while simultaneously upregulating 17β-HSD2 expression, thereby facilitating the conversion of estradiol into estrone, which has significantly lower estrogenic potency. This shift mitigates ERβ overstimulation in adipose tissue, a major driver of inflammation, fibrosis, and pathological expansion in lipedema [15].

Beyond its effects on estradiol metabolism, gestrinone also helps overcome progesterone resistance by stimulating the expression of PRβ receptors, restoring local hormonal balance. As a result, there is a reduction in chronic inflammation, fibrosis, and uncontrolled expansion of affected adipose tissue. Contrary to common misconceptions, its benefits in lipedema are not attributed to its androgenic or anabolic properties but rather to its capacity to reprogram aberrant intracrine steroid signaling, thus offering disease-modifying potential—especially in women with coexisting gynecological comorbidities [15,33]. Its greatest benefit appears to occur during the menopausal transition, particularly in women with overlapping gynecological conditions such as endometriosis, fibroids, and adenomyosis [37,38,39]. However, it is important to highlight that these potential therapeutic effects of gestrinone and drospirenone are primarily based on pathophysiological reasoning. No randomized clinical trials have yet evaluated their efficacy or safety in the context of lipedema. Despite theoretical benefits, the use of gestrinone and drospirenone in lipedema warrants caution. Gestrinone, for example, has been associated with androgenic side effects such as acne, hirsutism, and menstrual irregularities, while drospirenone carries a known risk of hyperkalemia and antiandrogenic effects.

In postmenopausal women, the use of micronized progesterone (P4) is a viable and well-established option. It plays a critical role in endometrial protection and neuroendocrine modulation. However, its impact on adipose tissue affected by lipedema is considered metabolically neutral, with no direct effect on the core pathophysiological mechanisms of the disease. Unlike synthetic progestins such as drospirenone and gestrinone, which modulate intracrine pathways, P4 does not significantly influence the expression or activity of enzymes like aromatase or 17β-HSD. Moreover, it lacks robust anti-adipogenic, antifibrotic, or anti-inflammatory effects in adipose tissue [19,25,40]. Although subcutaneous adipose tissue expresses progesterone receptors [19], activation of these receptors by natural progesterone is insufficient to reverse the pattern of progesterone resistance seen in lipedema or to modulate local estrogenic dysregulation. Therefore, while P4 is effective and safe for endometrial protection in hormone replacement therapy and beneficial for sleep regulation, anxiety, and neuroprotection, it does not offer a therapeutic impact on adipose inflammation, fibrosis, or hypertrophy—unlike drospirenone and gestrinone, which demonstrate specific actions on these metabolic pathways [35,40].

Metabolic therapy represents another fundamental pillar in the comprehensive management of lipedema, especially during the climacteric. Tirzepatide has emerged as a promising therapeutic strategy with high potential in this context [22], particularly when addressing the metabolic and hormonal alterations characteristic of midlife. Both lipedema and the climacteric share core pathophysiological mechanisms—such as chronic low-grade inflammation, tissue fibrosis, mitochondrial dysfunction, insulin resistance, and maladaptive remodeling of subcutaneous adipose tissue—that are exacerbated during the menopausal transition.

This transition further amplifies these mechanisms through systemic estradiol deficiency, intensified insulin resistance, impaired mitochondrial function, and increased central adiposity—creating a metabolically unfavorable environment that acts as a catalyst for the progression of lipedema [41].

In this context, tirzepatide—a dual GLP-1 and GIP receptor agonist—offers a multifaceted pharmacological profile that concurrently targets the metabolic and inflammatory dysfunctions common to both lipedema and the climacteric. Its anti-inflammatory properties are mediated through modulation of macrophage polarization, characterized by a reduction in pro-inflammatory M1 macrophages and an increase in anti-inflammatory M2 macrophages. Additionally, tirzepatide inhibits key inflammatory signaling pathways, including ERK and NF-κB, and downregulates the expression of pro-inflammatory cytokines such as TNF-α, IL-6, and MCP-1 [42,43].

Simultaneously, tirzepatide exerts antifibrotic effects and induces a metabolically favorable reprogramming relevant to both lipedema and the metabolic syndrome of menopause. Preclinical studies have demonstrated its capacity to stimulate thermogenesis via the upregulation of uncoupling protein 1 (UCP1) in brown adipose tissue, thereby enhancing energy expenditure and promoting the transdifferentiation of white adipocytes into beige adipocytes [44]. Furthermore, GIP receptor activation facilitates the mobilization of metabolically resistant subcutaneous fat—characteristic of lipedema—by improving mitochondrial function, increasing lipid oxidation, and regulating gene expression associated with energy homeostasis and extracellular matrix remodeling [44,45].

These mechanisms extend beyond mere weight reduction. By targeting the interrelated metabolic, inflammatory, and fibrotic pathways, tirzepatide delivers unique therapeutic advantages in the management of lipedema during the climacteric period—a condition where fat accumulation arises not solely from caloric excess but from complex endocrine, inflammatory, and bioenergetic dysfunctions driven by intracrine estrogen production, estrogen receptor imbalance, and progesterone resistance.

Compelling results from disease models such as metabolic-associated steatohepatitis (MASH) and heart failure with preserved ejection fraction (HFpEF) further support tirzepatide’s efficacy in reducing adipose tissue volume and attenuating processes of chronic inflammation, fibrosis, and mitochondrial dysfunction—key drivers in the pathogenesis of lipedema during menopause [46].

Therefore, by integrating metabolic, anti-inflammatory, antifibrotic, and tissue reprogramming effects, tirzepatide emerges as a physiopathologically informed and metabolically restorative intervention. It offers a novel therapeutic avenue capable of addressing the compounded clinical challenges posed by lipedema in the context of menopausal transition.

## 4. Material and Methods

This work is a narrative review aimed at developing an integrated pathophysiological model that explains how menopause acts as a critical turning point in the clinical progression of lipedema, with a focus on estrogen receptor imbalance, intracrine estrogen production, and adipose tissue dysfunction.

The selection of literature was based on a comprehensive search of the PubMed, Scopus, and Web of Science databases up to May 2025. Keywords used included combinations of “lipedema”, “estrogen receptors”, “ERα”, “ERβ”, “intracrine estrogen”, “adipose tissue dysfunction”, “menopause”, “aromatase”, “17β-HSD”, “progesterone resistance”, “fibrosis”, and “inflammation”. No language or date restrictions were applied to maximize the retrieval of relevant studies.

The literature included original research articles, molecular studies, clinical studies, and previous reviews that specifically addressed the molecular biology of estrogen signaling, adipose tissue endocrinology, and gynecological hormone-sensitive disorders (e.g., endometriosis, adenomyosis, and uterine fibroids) in relation to lipedema.

Articles were selected based on their relevance to the following domains:Estrogen receptor signaling (ERα, ERβ) in adipose tissue;Intracrine estrogen metabolism via aromatase, 17β-HSD1, and 17β-HSD2 enzymes;The role of menopause-induced estrogen deficiency in adipose tissue dysfunction;The immunometabolic consequences of receptor imbalance, including inflammation and fibrosis;Parallels between lipedema and other estrogen-driven gynecological disorders.

This research combines mechanistic insights derived from molecular endocrinology, adipose tissue biology, and gynecological studies to propose a new conceptual framework and advance the scientific understanding of lipedema as a hormonal disorder, particularly in the context of menopausal changes.

## 5. Conclusions

This narrative review, despite inherent limitations—particularly its reliance on pathophysiological models and observational data without validation through randomized clinical trials specific to lipedema—provides an expanded and coherent understanding of the mechanisms that position menopause as a critical inflection point in the clinical course of the disease.

The systemic decline in circulating estradiol, coupled with exacerbated intracrine estradiol production in affected adipose tissue—driven by aromatase and 17β-HSD1 overexpression and 17β-HSD2 deficiency—along with an imbalance favoring ERβ over ERα signaling, establishes a pro-inflammatory, profibrotic, and estrogen-dominant microenvironment. This hormonal milieu promotes adipocyte hypertrophy, chronic inflammation, extracellular matrix remodeling, and resistance to lipid mobilization.

Beyond hormonal mechanisms, menopause induces profound metabolic changes—including increased insulin resistance, chronic low-grade inflammation, mitochondrial dysfunction, and redistribution of adiposity to the visceral compartment. These events synergize with the pathophysiology of lipedema, amplifying disease severity and contributing to a more aggressive clinical phenotype during the climacteric.

In this context, the need for a strategic therapeutic shift becomes evident. Transdermal estradiol—applied to unaffected areas such as the back or upper arms—emerges as a central intervention to restore the ERα/ERβ balance, suppress inflammatory cascades, and prevent fibrotic remodeling of adipose tissue.

Equally relevant is the use of progestins with anti-inflammatory and metabolic actions, such as drospirenone and gestrinone. These agents extend beyond endometrial protection by actively modulating progesterone resistance and rebalancing intracrine estrogen synthesis through downregulation of aromatase and 17β-HSD1 and upregulation of 17β-HSD2, thereby mitigating key mechanisms sustaining lipedema progression.

Complementing the endocrine approach, tirzepatide represents a novel and promising metabolic therapy. Its dual agonism of GLP-1 and GIP receptors provides multiple synergistic actions: reduction of insulin resistance, enhancement of mitochondrial function, stimulation of thermogenesis and energy expenditure, attenuation of chronic inflammation and fibrosis, and direct remodeling of dysfunctional adipose tissue. These effects uniquely address the complex hormonal–metabolic–inflammatory triad that characterizes climacteric lipedema.

Unlike GLP-1 analogs alone, tirzepatide’s dual action enables not only weight reduction but also partial reversal of adipose tissue dysfunction, particularly when compounded by menopausal endocrine disruption.

Thus, the recognition of menopause as a pivotal pathophysiological milestone in lipedema progression not only enhances our understanding of disease mechanisms but also redefines therapeutic priorities. The integration of personalized hormonal strategies (transdermal estradiol and selective progestins) with advanced metabolic interventions such as tirzepatide offers a physiologically grounded and individualized approach with the potential to transform the clinical management of lipedema in menopausal women.

This review is anchored in an integrative pathophysiological model derived from observational studies, molecular biology, and mechanistic extrapolations from related estrogen-dependent disorders. While biologically plausible and supported by a robust mechanistic rationale, the absence of randomized controlled trials specifically addressing lipedema populations remains a significant limitation. Future research should focus on validating these mechanisms and translating the proposed therapeutic interventions into evidence-based clinical protocols.

## Figures and Tables

**Figure 1 ijms-26-07074-f001:**
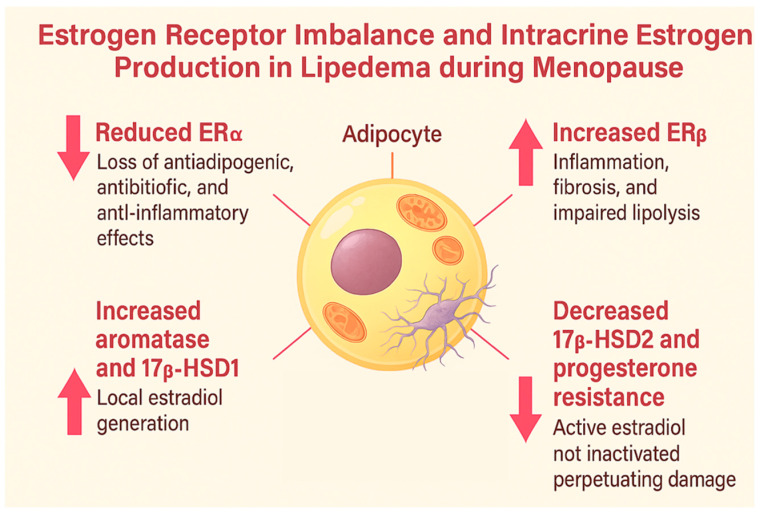
Estrogen receptor imbalance and intracrine estrogen production in lipedema during menopause. Reduced ERα impairs anti-inflammatory and antiadipogenic effects, while increased ERβ promotes inflammation, fibrosis, and impaired lipolysis. Local estradiol synthesis is intensified by aromatase and 17β-HSD1, and its inactivation is reduced due to decreased 17β-HSD2 and progesterone resistance, leading to sustained estrogenic activation.

**Figure 2 ijms-26-07074-f002:**
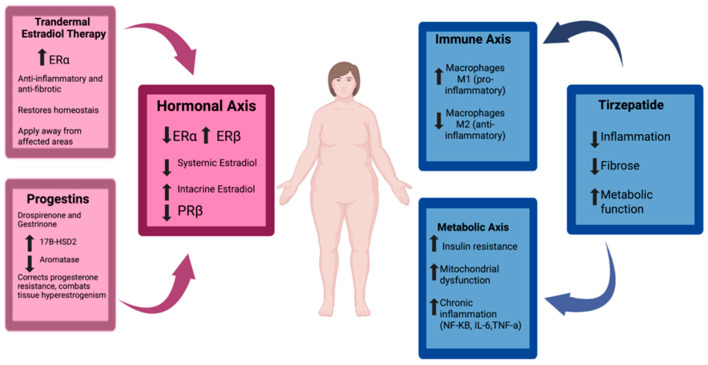
Mechanisms by which menopause worsens lipedema. Menopause induces hormonal, immune, and metabolic changes—such as ERβ dominance, progesterone resistance, and chronic inflammation—that exacerbate lipedema. The figure illustrates how these disruptions worsen fibrosis, fat accumulation, and treatment resistance, andit highlights potential therapeutic targets like estradiol, progestins, and tirzepatide. Created in BioRender. Viana, D. (2025) https://BioRender.com/7wicoqn.

**Table 1 ijms-26-07074-t001:** Key pathophysiological alterations in lipedema and menopause-related adipose tissue dysfunction. This table summarizes the main molecular and hormonal changes associated with lipedema, highlighting the roles of estrogen receptors (ERα/ERβ), steroidogenic enzymes (aromatase, 17β-HSD1/2), progesterone resistance, and mitochondrial dysfunction. Each mechanism is supported by recent literature to illustrate its contribution to inflammation, fibrosis, and metabolic impairment in lipedema.

Pathophysiological Change	Author and Year	Summary of Findings
Estrogen Receptors (ERα and ERβ)	Simpson et al., 1989 [8]	The climacteric period decreases ERα expression and compensatorily increases ERβ in adipose tissue.
	Foryst-Ludwig and Kintscher, 2010 [13]	Estrogen deficiency in menopause and lipedema reduces ERα and increases ERβ, promoting inflammation, fibrosis, and insulin resistance.
	Katzer K et al., 2021 [4]	Dysregulation of estrogen receptors (ERα/ERβ) and local estrogen production in adipose tissue may lead to excessive fat accumulation, particularly in the lower body, a hallmark of lipedema.
Aromatase (CYP19A1)	Simpson et al., 1989 [8]	Subcutaneous adipose tissue synthesizes estrogens via aromatase and 17β-HSD.
	Szél et al., 2014 [16]	Lipedema exhibits increased aromatase activity and enzymatic dysregulation.
17β-HSD 1 and 2	Zeitoun et al., 1998 [19]	In conditions like endometriosis, 17β-HSD2 deficiency prevents the conversion of estradiol into estrone.
	Szél et al., 2014 [16]	There is an increase in 17β-HSD1, which converts estrone into active estradiol, intensifying local estrogenic activation.
	Bardhi et al., 2024 [20]	17β-HSDs in adipose tissue convert weak steroids like estrone into potent forms such as estradiol, underscoring the role of local hormone metabolism in adipose function.
	Al-Ghadban et al., 2024 [21]	Estrogen enhances HSD17B7 and LIPE expression in lipedema cells, supporting a direct role of estrogen metabolism in disease pathogenesis.
	Viana and Câmara, 2025 [22]	Progesterone resistance reduces 17β-HSD2 activity, impairing estradiol inactivation.
Progesterone Resistance	O’Brien et al., 1998 [18]	Subcutaneous adipose tissue expresses progesterone receptors, suggesting an active local hormonal role.
	Viana and Câmara, 2025 [15]	The failure of progesterone to modulate adipose tissue allows intracrine estradiol to sustain the inflammatory state.
Mitochondrial Dysfunction	Geraci et al., 2021 [23] Renke et al., 2023 [24]	Mitochondrial dysfunction induced by estrogen deficiency reduces basal metabolism, contributing to sarcopenia and insulin resistance.

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
