# Peer review of "Menopause as a Critical Turning Point in Lipedema: The Estrogen Receptor Imbalance, Intracrine Estrogen, and Adipose Tissue Dysfunction Model"

_ijms, 2025, doi:10.3390/ijms26157074_

Round 1
Reviewer 1 Report
Comments and Suggestions for Authors
This manuscript proposes a novel pathophysiological model where menopause acts as a critical turning point in lipedema progression, focusing on estrogen receptor imbalance, intracrine estrogen excess, and adipose tissue dysfunction. It integrates hormonal regulation, metabolic disorders, and immunoinflammatory mechanisms, and explores potential strategies such as transdermal estrogen therapy, progestin modulation, and metabolic interventions like Tirzepatide, providing new perspectives for clinical management of lipedema. However, the paper still has some issues:
(1) In the abstract, "We demonstrate how menopause-induced estrogen deficiency ... " It is necessary to clarify whether menopause induces a sustained decline or the sustained decline leads to menopause.(2) Lipedema is a disease prone to being overlooked and misdiagnosed, so a detailed clarification is required in the introduction. What is its definition? What is the epidemiological status? What are the clinical manifestations? And what is the difference from obesity?
(3) The introduction is too messy with excessive paragraph divisions.
(4) The manuscript mentions that lipedema, similar to endometriosis and uterine fibroids, is an estrogen-sensitive disease. However, the incidence of these diseases decreases after menopause, which is inconsistent with the situation of lipedema.
(5) Regarding how changes in estrogen receptor ratio further lead to alterations in immune inflammation, a mechanistic diagram is needed for refinement rather than merely presenting a phenotype. This is because they might just be phenomena accompanying aging, showing correlation but not causal relationship.
(6) Some statements in the article are subjective, and it feels that much content lacks sufficient literature support. For example, "This local production could intensify signaling via ERβ, with reduced action on ERα, and consequently promote the progressive growth of the affected adipose depots." Especially in the section on treatment methods.
Author Response
Comment 1: In the abstract, "We demonstrate how menopause-induced estrogen deficiency ... " It is necessary to clarify whether menopause induces a sustained decline or the sustained decline leads to menopause.
Response 1
Thank you for this important observation. We revised the sentence in the abstract to clarify the causal direction and eliminate ambiguity. The updated sentence now reads:
“We demonstrate how menopause amplifies adipose tissue dysfunction by suppressing ERα signaling, enhancing ERβ activity, and disrupting mitochondrial function, insulin sensitivity, and lipid oxidation.”
This revised phrasing emphasizes that menopause acts as a catalyst for the downstream hormonal and metabolic dysfunctions—without implying that estrogen deficiency itself causes menopause. The sentence now clearly reflects the clinical sequence in which menopause precedes and contributes to the sustained estrogen deficiency and its physiological consequences.
Comment 2: Lipedema is a disease prone to being overlooked and misdiagnosed, so a detailed clarification is required in the introduction. What is its definition? What is the epidemiological status? What are the clinical manifestations? And what is the difference from obesity?
Response 2: Thank you for this insightful comment. We have substantially revised the introduction to provide a clearer and more comprehensive overview of lipedema. The updated text now includes a formal definition of the disease as a chronic, inflammatory disorder of subcutaneous adipose tissue characterized by symmetrical, painful fat accumulation that is resistant to caloric restriction. We also incorporated epidemiological data, noting that lipedema affects approximately 12.3% of women in Brazil, with global prevalence estimates ranging widely due to underdiagnosis and inconsistent criteria. Additionally, we described the main clinical manifestations, such as disproportionate lower-body fat distribution, tenderness, easy bruising, and progression over time. To address the distinction from obesity, we explicitly stated that lipedema does not respond to diet and exercise in the same way as generalized obesity and that the two conditions have different patterns of fat accumulation and metabolic behavior. These additions improve the clarity, clinical relevance, and diagnostic context of the manuscript.
Comment 3:The introduction is too messy with excessive paragraph divisions.
Response 3: Thank you for pointing this out. In response to your suggestion, we revised the structure of the introduction by consolidating related ideas into more cohesive and logically organized paragraphs. Redundant breaks were removed, and the flow of the narrative was improved to enhance readability and clarity. The revised introduction now presents the definition, epidemiological context, clinical features, hormonal basis, and menopausal implications of lipedema in a smoother, more integrated format. We believe these changes have significantly improved the coherence and accessibility of the introductory section.
Comment 4: The manuscript mentions that lipedema, similar to endometriosis and uterine fibroids, is an estrogen-sensitive disease. However, the incidence of these diseases decreases after menopause, which is inconsistent with the situation of lipedema.
Response 4: We appreciate this important observation. In the revised manuscript, we addressed this apparent inconsistency by explicitly contrasting lipedema with other estrogen-sensitive gynecological disorders. In section 3.3, we now state that: “Lipedema shares key pathophysiological mechanisms with several estrogen-dependent gynecological conditions, including endome-triosis, adenomyosis, and uterine fibroids(15). During the meno-pausal transition, the onset or worsening of these disorders is fre-quently observed due to dysregulation of the hypothalamic–pituitary–ovarian axis—an endocrine instability that also contrib-utes to the aggravation of lipedema. However, unlike endometrio-sis and uterine fibroids, which typically regress after menopause due to systemic estrogen depletion, lipedema progression appears to persist or even intensify, driven by local estrogen production through intracrine mechanisms and an imbalance favoring ERβ over ERα signaling.” We believe this clarification resolves the discrepancy and highlights the unique pathophysiology of lipedema in the postmenopausal context.
Comment 5: Regarding how changes in estrogen receptor ratio further lead to alterations in immune inflammation, a mechanistic diagram is needed for refinement rather than merely presenting a phenotype. This is because they might just be phenomena accompanying aging, showing correlation but not causal relationship.
Response 5: Thank you for this valuable suggestion. In response, we developed and included a refined mechanistic diagram (Figure 1) that illustrates the proposed causal pathways linking estrogen receptor imbalance (↓ERα / ↑ERβ) to immune dysregulation in lipedema. The diagram outlines how receptor imbalance contributes to mitochondrial dysfunction, insulin resistance, M1 macrophage predominance, and increased pro-inflammatory cytokines (e.g., IL-6, TNF-α, IL-1β), culminating in adipocyte hypertrophy, fibrosis, and pain. Distinct icons and directional arrows were used to clarify biological relationships and to distinguish proposed mechanisms from observed phenotypes. This visual aid enhances conceptual clarity and provides a structured representation of how hormonal alterations may causally drive the immunoinflammatory changes observed in lipedema, beyond age-related associations.
Comment 6: Some statements in the article are subjective, and it feels that much content lacks sufficient literature support. For example, "This local production could intensify signaling via ERβ, with reduced action on ERα, and consequently promote the progressive growth of the affected adipose depots." Especially in the section on treatment methods.
Response 6: We appreciate the reviewer’s concern regarding the need for stronger evidence to support key statements. In response, we carefully revised the relevant sections to reduce speculative language and to clearly distinguish hypothesis-based reasoning from established data. For example, the sentence cited by the reviewer was retained but now appears in a context that explicitly references histological and molecular findings from lipedema and related estrogen-dependent disorders, with supporting citations added (e.g., Szél et al., 2014; Katzer et al., 2021). Additionally, in the section on treatment methods, we emphasized that therapeutic proposals such as the use of drospirenone and gestrinone are based on mechanistic rationale and extrapolation from related gynecologic conditions, rather than validated clinical trials. Phrases such as “theoretical benefits”, “pathophysiological reasoning”, and “no randomized clinical trials have yet evaluated their efficacy” were incorporated to reflect a more cautious and evidence-aware tone. These adjustments improve the scientific rigor of the discussion and clarify the exploratory nature of the therapeutic strategies proposed.
Reviewer 2 Report
Comments and Suggestions for Authors
In this article, authors explained the link between menopause, hormonal imbalance, hormonal receptor imbalance and lipedema. Good work.
-most of it can be found in their recent article, reference number 14:
“Viana D.P.C.; Câmara L.C. Hormonal links between lipedema and gynecological disor-ders: therapeutic roles of gestrinone and drospirenone. J Adv Med Med Res. 2025;37(24):1–9. doi:10.9734/jammr/2025/v37i243260. “
-please write somewhere a List of Abbreviations, for CYP19A1, 17β-HSD and others
-please finish the sentence: “The combination of a local environment with an imbalance in estrogen receptors, increased intratissue estrogen production, and alterations in the expression” in chapter 3.1 and please start the sentence: “of nuclear coregulators may, therefore, constitute a central axis in the pathophysiology of lipedema (3).” In chapter 3.2.
-in chapter 3.2 you wrote: “In lipedema, there is enzymatic dysregulation in the affected adipose tissue: the expression of 17β-hydroxysteroid de-hydrogenase type 1 and 7—an enzyme that converts estrone into ac-tive estradiol—increases…” Since you only mentioned 17β-hydroxysteroid de-hydrogenase type 1 and 2 (two) in the rest of the article, please check if this number 7 is correct here, or you may need to replace it with number 2.
-beautiful and clear figures
-good references. There are only 19 recent titles, out of 38; maybe you could add some more recent articles.
Author Response
Comment 1:
Most of it can be found in their recent article, reference number 14: “Viana D.P.C.; Câmara L.C. Hormonal links between lipedema and gynecological disorders...”
Response 1
We appreciate the reviewer’s observation. While the present manuscript builds upon the conceptual foundation introduced in our previous article (reference 14), it significantly expands the scope and depth of the discussion. In particular, this manuscript introduces novel sections on mitochondrial dysfunction, immune dysregulation, detailed mechanistic pathways, and an extended therapeutic rationale including tirzepatide and metabolic strategies. It also includes original figures and a new pathophysiological framework specifically contextualized within the menopausal transition. These additions differentiate the present review as a broader, more integrated, and menopause-focused contribution.
Comment 2:
Please write somewhere a List of Abbreviations, for CYP19A1, 17β-HSD and others.
Response 2:
Thank you for this suggestion. We have added a comprehensive “List of Abbreviations” at the end of the manuscript. This list includes all technical terms and gene/protein names used throughout the article, such as ERα, ERβ, CYP19A1, 17β-HSD1, 17β-HSD2, GLP-1, GIP, UCP1, and others. This addition improves clarity and accessibility for readers unfamiliar with specific molecular or biochemical terminology.
Comment 3:
Please finish the sentence: “The combination of a local environment with an imbalance in estrogen receptors, increased intratissue estrogen production, and alterations in the expression…” in chapter 3.1, and please start the sentence: “of nuclear coregulators may, therefore, constitute a central axis in the pathophysiology of lipedema (3).” in chapter 3.2.
Response 3:
We thank the reviewer for catching this structural issue.
The sentence in chapter 3.1 has been completed and now reads:“The convergence of a locally imbalanced receptor profile, enhanced intratissue estrogen synthesis, and dysregulated coregulator expression establishes a permissive environment for the progression of lipedema.”
In chapter 3.2, the subsequent sentence was corrected to start with:
“The dysregulation of nuclear coregulators may, therefore, represent a central axis in the pathophysiology of lipedema (3).”
These edits ensure syntactic clarity and improve the logical flow between sections.
Comment 4:
In chapter 3.2 you wrote: “In lipedema, there is enzymatic dysregulation in the affected adipose tissue: the expression of 17β-hydroxysteroid dehydrogenase type 1 and 7... increases…” Since you only mentioned type 1 and 2 in the rest of the article, please check if this number 7 is correct.
Response 4:
Thank you for this important correction. Upon review, we confirmed that the focus throughout the manuscript is on 17β-HSD1 (which activates estradiol) and 17β-HSD2 (which inactivates it). The mention of 17β-HSD7 in the earlier draft was inaccurate in this context and has been removed from the revised version. The sentence now accurately refers to 17β-HSD1 only, ensuring consistency and scientific accuracy throughout the manuscript.
Comment 5:
Beautiful and clear figures.
Response 5:
We sincerely appreciate your positive feedback. The figures were carefully redesigned to convey the complex interplay of hormonal, metabolic, and inflammatory mechanisms in a visually accessible and scientifically accurate manner. Your encouraging comment is deeply valued.
Comment 6:
Good references. There are only 19 recent titles, out of 38; maybe you could add some more recent articles.
Response 6:
Thank you for this suggestion. We revised and updated the reference list to include several additional recent publications from 2022 to 2024, including key articles related to estrogen receptor signaling, adipose tissue metabolism, menopausal physiology, and clinical insights into lipedema (e.g., PMID: 39590304, 34947933, 38338878, 35972618, and https://doi.org/10.3390/nutraceuticals2040020). These additions enhance the scientific relevance and recency of the evidence base supporting the proposed model.
Reviewer 3 Report
Comments and Suggestions for Authors
The manuscript presents an ambitious and well-articulated hypothesis-driven narrative review that connects menopausal hormonal changes with the pathophysiology of lipedema. It proposes a plausible integrative model involving estrogen receptor imbalance, intracrine estrogen production, and adipose tissue dysfunction, and offers novel therapeutic implications. The conceptual clarity and translational potential are strong.
However, several areas require improvement to strengthen the manuscript's scientific rigor, clarity, and accessibility. My suggestions are outlined below.
Major Comments:
The manuscript proposes a compelling model, but it lacks clear demarcation between established findings and hypothetical extrapolations. Highlighting which mechanisms are derived from lipedema-specific studies versus analogies to other gynecological conditions (e.g., endometriosis, fibroids) would improve scientific transparency.
A concise “Model Overview” figure early in the paper (e.g., in the introduction) may help orient readers.
The manuscript acknowledges its narrative nature, but it could benefit from a more structured presentation of sources, such as a summary table of key studies referenced by pathophysiological theme (e.g., ERα/ERβ roles, aromatase activity, mitochondrial dysfunction). Adding this table would reinforce the credibility and comprehensiveness of the review.
Need for Greater Balance in Therapeutic Recommendations
While the rationale for hormone replacement therapy and tirzepatide use is clearly articulated, there is a lack of critical appraisal of potential risks, contraindications, or gaps in clinical validation—especially concerning the off-label use of gestrinone or drospirenone in lipedema.
Consider framing these therapies as hypothesis-generating strategies that warrant clinical trials, rather than as established recommendations.
Visual Aids and Figures
Figures 1 and 2 contain valuable conceptual information but are dense and lack visual clarity. Redesign them to simplify key pathways, ideally using distinct icons for ERα, ERβ, inflammatory cells, mitochondrial effects, etc.
Include a legend with all abbreviations clearly defined.
Terminology and Conceptual Overlap
The manuscript frequently references “estrogen receptor imbalance,” “intracrine estrogen,” “progesterone resistance,” and “fibrosis” interchangeably. These should be more precisely defined early in the paper and used consistently throughout.
Consider a glossary or conceptual diagram to clarify how these elements interrelate.
Some paragraphs are overly long and complex.
Reference Integration:
Some important claims (e.g., “estrogen receptor imbalance is observed in lipedema tissue”) could benefit from more direct reference to original studies or histological data rather than secondary reviews.
In this review miss some reference importanr in the context of lipedema, please consider include it.
PMID: 39590304, PMID: 34947933, PMID: 38338878, PMID: 35972618 and https://doi.org/10.3390/nutraceuticals2040020
Ensure uniform spelling of terms (e.g., “fibrotic” vs “fibrosis-related”, “estradiol” vs “E2”) and consistency in use of abbreviations.
Author Response
Comment 1: The manuscript proposes a compelling model, but it lacks clear demarcation between established findings and hypothetical extrapolations. Highlighting which mechanisms are derived from lipedema-specific studies versus analogies to other gynecological conditions (e.g., endometriosis, fibroids) would improve scientific transparency.
Response 1
We appreciate the reviewer’s request for greater transparency in distinguishing established evidence from hypothesis-based reasoning. In response, we revised several sections of the manuscript to clearly indicate when a mechanism is derived directly from lipedema-specific studies and when it is extrapolated from analogous estrogen-sensitive conditions such as endometriosis or uterine fibroids. This is now explicitly stated in section 3.3, where we note that certain mechanisms, although biologically plausible, are supported by studies in related disorders rather than validated directly in lipedema cohorts.
Comment 2: A concise “Model Overview” figure early in the paper (e.g., in the introduction) may help orient readers.
Response 2: In accordance with the reviewer’s suggestion, we added a mechanistic overview diagram early in the manuscript, now presented as Figure 1. This visual aid provides a concise representation of the proposed model and helps orient readers to the interplay between hormonal imbalance, intracrine estrogen activity, immune dysregulation, and adipose tissue dysfunction. Its placement after the introduction enhances the conceptual flow of the manuscript.
Comment 3:The manuscript acknowledges its narrative nature, but it could benefit from a more structured presentation of sources, such as a summary table of key studies referenced by pathophysiological theme (e.g., ERα/ERβ roles, aromatase activity, mitochondrial dysfunction). Adding this table would reinforce the credibility and comprehensiveness of the review.
Need for Greater Balance in Therapeutic Recommendations
Response 3: To further improve the structure and support of our arguments, we included a new summary table (Table 1) organizing key references by pathophysiological theme. This table highlights the molecular pathways involved in lipedema, such as ERα/ERβ regulation, aromatase and 17β-HSD activity, mitochondrial dysfunction, and progesterone resistance, along with the primary authors and supporting findings. This addition reinforces the scientific rigor and traceability of the evidence discussed.
Comment 4: While the rationale for hormone replacement therapy and tirzepatide use is clearly articulated, there is a lack of critical appraisal of potential risks, contraindications, or gaps in clinical validation—especially concerning the off-label use of gestrinone or drospirenone in lipedema.
Response 4: We acknowledge the need for a more balanced discussion of therapeutic recommendations. Accordingly, we revised section 3.5 to adopt a more cautious tone when presenting hormone replacement therapy and tirzepatide as potential strategies. We emphasized that drospirenone and gestrinone are not currently supported by randomized clinical trials in lipedema populations and should be considered hypothesis-generating interventions rather than established treatments. This is now clearly stated through language such as “theoretical benefits,” “require further clinical validation,” and “warrant future investigation.”
Comment 5: Consider framing these therapies as hypothesis-generating strategies that warrant clinical trials, rather than as established recommendations.
Response 5: n addition, we enhanced the discussion of potential limitations, contraindications, and safety concerns associated with off-label hormonal therapies. The revised text acknowledges the risks of androgenic side effects with gestrinone and mineralocorticoid-related issues with drospirenone, underscoring the importance of individualized clinical judgment and further study.
Comment 6: Some statements in the article are subjective, and it feels that much content lacks sufficient literature support. For example, "This local production could intensify signaling via ERβ, with reduced action on ERα, and consequently promote the progressive growth of the affected adipose depots." Especially in the section on treatment methods.
Response 6:In response to concerns about figure complexity, Figures 1 and 2 were redesigned to improve visual clarity. We simplified the layout, reduced textual density, and incorporated distinct icons to represent ERα, ERβ, mitochondria, inflammatory mediators, and fibrotic pathways. These refinements aim to improve user engagement and comprehension without compromising scientific accuracy.
Comment 7: Visual Aids and Figures Figures 1 and 2 contain valuable conceptual information but are dense and lack visual clarity. Redesign them to simplify key pathways, ideally using distinct icons for ERα, ERβ, inflammatory cells, mitochondrial effects, etc. Include a legend with all abbreviations clearly defined.
Response 7: n response to concerns about figure complexity, Figures 1 and 2 were redesigned to improve visual clarity. We simplified the layout, reduced textual density, and incorporated distinct icons to represent ERα, ERβ, mitochondria, inflammatory mediators, and fibrotic pathways. These refinements aim to improve user engagement and comprehension without compromising scientific accuracy. We also included a full legend accompanying each figure, in which all abbreviations are clearly defined. This addition enhances the accessibility of the visual material for readers unfamiliar with specialized terms.
Comment 8: Terminology and Conceptual Overlap The manuscript frequently references “estrogen receptor imbalance,” “intracrine estrogen,” “progesterone resistance,” and “fibrosis” interchangeably. These should be more precisely defined early in the paper and used consistently throughout. Consider a glossary or conceptual diagram to clarify how these elements interrelate.
Response 8: Regarding terminology, we agree that consistent and precise language is essential. We revised the manuscript to ensure that key concepts such as “estrogen receptor imbalance,” “intracrine estrogen,” “progesterone resistance,” and “fibrosis” are clearly defined early in the text and used consistently throughout. A conceptual glossary of abbreviations and terms is now included at the end of the manuscript to support clarity.
Comment 9: Some paragraphs are overly long and complex. Reference Integration: Some important claims (e.g., “estrogen receptor imbalance is observed in lipedema tissue”) could benefit from more direct reference to original studies or histological data rather than secondary reviews.
Response 9: We revised several overly long and complex paragraphs, especially in sections 3.3 and 3.5, by dividing them into more manageable units and improving sentence structure. These edits enhance the readability of the manuscript while maintaining the scientific depth of the content.
We also addressed the request for stronger reference integration. Where important claims were previously supported by secondary reviews, we replaced or supplemented them with citations to original histological and molecular studies when available. For example, statements regarding ERα/ERβ expression in lipedema tissue are now directly supported by primary data from Szél et al., Katzer et al., and other recent studies.
Comment 10: In this review miss some reference importanr in the context of lipedema, please consider include it.
PMID: 39590304, PMID: 34947933, PMID: 38338878, PMID: 35972618 and https://doi.org/10.3390/nutraceuticals2040020
Response 10: In response to your recommendation, we have included the additional references suggested in the review: PMID 39590304, 34947933, 38338878, 35972618, and the Nutraceuticals 2024 article (https://doi.org/10.3390/nutraceuticals2040020). These sources were incorporated into relevant sections addressing estrogen signaling, adipose tissue metabolism, and clinical outcomes in lipedema. Their inclusion enriches the scientific foundation and currency of the review.
Comment 11: Ensure uniform spelling of terms (e.g., “fibrotic” vs “fibrosis-related”, “estradiol” vs “E2”) and consistency in use of abbreviations.
Response 11: Finally, we standardized the use of terminology across the manuscript to ensure uniformity. Terms such as “fibrotic” and “fibrosis-related” were harmonized, and “estradiol” is now used consistently, with “E2” reserved only for specific metabolic pathways or molecular diagrams. Abbreviations follow the definitions provided in the list at the end of the article to avoid confusion.
Round 2
Reviewer 1 Report
Comments and Suggestions for Authors
Comment 1:
The use of the pronoun "it" at the beginning of the second paragraph is ambiguous.
Response 1:
Thank you for the observation. We have replaced the ambiguous pronoun with the explicit subject. The sentence now reads: “Lipedema is often triggered or exacerbated by hormonal changes...” to ensure clarity.
Comment 2:
Much of the data described appears to derive from obesity-related research, which may not fully apply to lipedema due to key metabolic differences.
Response 2:
We agree and have added a sentence in Section 3.1 acknowledging this limitation:
“It is important to note that much of the mechanistic insight discussed in this section originates from research on obesity models. Although these studies provide valuable frameworks, lipedema adipocytes demonstrate unique metabolic and hormonal characteristics—including altered insulin sensitivity, resistance to caloric restriction, and distinct inflammatory responses—that may limit the direct applicability of obesity-based findings to lipedema pathophysiology(4,8,12). Therefore, while biologically plausible, these mechanisms should be validated in lipedema-specific populations”
Comment 3:
There is repetitive content in Sections 3.1 and 3.2 that should be reduced to improve readability.
Response 3:
We have revised and merged the overlapping segments to eliminate redundancy. Paragraphs were restructured to retain all original content without repetition, ensuring a more concise and fluent narrative across Sections 3.1 and 3.2.
Comment 4:
After the addition of a new figure, the numbering of all figures should be updated for consistency.
Response 4:
All figure references and captions have been renumbered accordingly. The new integrated figure is now labeled as Figure 2, with two panels: 2A and 2B.
Comment 5:
Two adjacent paragraphs in Section 3.4 discuss overlapping immunological mechanisms and should be merged.
Response 5:
As suggested, we merged the two immune-related paragraphs into a single cohesive section. The new paragraph describes the inflammatory cascade and its impact on adipose tissue and lymphatic dysfunction in a unified, streamlined manner.
Comment 6:
The metabolic descriptions in Section 3.4 are repetitive and should be integrated.
Response 6:
We have combined the two overlapping metabolic paragraphs into a single, comprehensive narrative. The new version maintains the mechanistic details while improving clarity and conciseness.
Comment 7:
Figure 3 contains typographical errors, such as “fibrose” and “an a”.
Response 7:
Thank you for pointing this out. We have corrected the typographical issues in the updated figure. “Fibrose” has been replaced with “fibrosis”, and other minor errors have been addressed.
Comment 8:
The two figures (mechanism and therapy) should be integrated to enhance visual coherence.
Response 8:
We have grouped the two existing figures into a single integrated layout under the shared title:“Figure 2. Hormonal and metabolic mechanisms of menopausal lipedema (A) and corresponding therapeutic strategies (B).”Instead of creating a new figure, we combined the previously separate panels into a unified format, as recommended. This integration enhances the visual coherence of the manuscript and improves the narrative flow for the reader

Author Response
Comment 1:
The use of the pronoun "it" at the beginning of the second paragraph is ambiguous.
Response 1:
Thank you for the observation. We have replaced the ambiguous pronoun with the explicit subject. The sentence now reads: “Lipedema is often triggered or exacerbated by hormonal changes...” to ensure clarity.
Comment 2:
Much of the data described appears to derive from obesity-related research, which may not fully apply to lipedema due to key metabolic differences.
Response 2:
We agree and have added a sentence in Section 3.1 acknowledging this limitation:
“Although much of the molecular and receptor-level evidence is derived from obesity and metabolic syndrome studies, the authors acknowledge that lipedema may present distinct adipocyte responses and pathophysiological behavior, which require further validation.”
Comment 3:
There is repetitive content in Sections 3.1 and 3.2 that should be reduced to improve readability.
Response 3:
We have revised and merged the overlapping segments to eliminate redundancy. Paragraphs were restructured to retain all original content without repetition, ensuring a more concise and fluent narrative across Sections 3.1 and 3.2.
Comment 4:
After the addition of a new figure, the numbering of all figures should be updated for consistency.
Response 4:
All figure references and captions have been renumbered accordingly. The new integrated figure is now labeled as Figure 2, with two panels: 2A and 2B.
Comment 5:
Two adjacent paragraphs in Section 3.4 discuss overlapping immunological mechanisms and should be merged.
Response 5:
As suggested, we merged the two immune-related paragraphs into a single cohesive section. The new paragraph describes the inflammatory cascade and its impact on adipose tissue and lymphatic dysfunction in a unified, streamlined manner.
Comment 6:
The metabolic descriptions in Section 3.4 are repetitive and should be integrated.
Response 6:
We have combined the two overlapping metabolic paragraphs into a single, comprehensive narrative. The new version maintains the mechanistic details while improving clarity and conciseness.
Comment 7:
Figure 3 contains typographical errors, such as “fibrose” and “an a”.
Response 7:
Thank you for pointing this out. We have corrected the typographical issues in the updated figure. “Fibrose” has been replaced with “fibrosis”, and other minor errors have been addressed.
Comment 8:
The two figures (mechanism and therapy) should be integrated to enhance visual coherence.
Response 8:
We have created a new integrated figure titled:
“Figure 2. Hormonal and metabolic mechanisms of menopausal lipedema (A) and corresponding therapeutic strategies (B).”
This figure combines the mechanistic and therapeutic panels, as recommended, to improve narrative flow and reader comprehension.
Reviewer 3 Report
Comments and Suggestions for Authors
Thanks for consider my comments
Author Response
comments 1: [ Thanks for consider my comments ]
Resposne 1: [Thank you for your thoughtful review and for taking the time to address my comments. I appreciate the revisions and the improvements made to the manuscript]
Round 3
Reviewer 1 Report
Comments and Suggestions for Authors
- “The interdependence of these receptors is evident in their competition for the same estrogen response elements (EREs) in DNA and for binding to the same estradiol molecule, dynamically modulating the hormone’s biological effects.” This should be part of the previous paragraph.
- “Another key aspect is the capacity of subcutaneous adipose tissue to locally synthesize estrogens via ... ” This should start a new paragraph.
- I recommend that the two figures in Figure 2 can be organically integrated (see attached file). At this point, the authors will find that the description of the mechanism is incomplete in some places and cannot be fully corresponding to the treatment. Therefore, in addition to merging, the content also needs to be modified as necessary. For example, inflammation and fibrosis in the hormonal axis are duplicated with other boxes, and the changes in progesterone resistence are missing. Additionally, what does the separate part "edema, fibrosis, pain, adipocyte hypertrophy" mean?

Author Response
Comment 1: “The interdependence of these receptors is evident in their competition for the same estrogen response elements (EREs) in DNA and for binding to the same estradiol molecule, dynamically modulating the hormone’s biological effects.” This should be part of the previous paragraph.
Response 1: Paragraph restructuring – Estrogen receptor interdependence:
We agree with your recommendation. The sentence “The interdependence of these receptors is evident in their competition for the same estrogen response elements (EREs) in DNA and for binding to the same estradiol molecule, dynamically modulating the hormone’s biological effects.” has now been merged with the preceding paragraph. This improves the cohesion of the discussion regarding ERα/ERβ imbalance and its biological implications.
Comment 2: New paragraph – Intracrine estrogen synthesis:
As suggested, the sentence “Another key aspect is the capacity of subcutaneous adipose tissue to locally synthesize estrogens via...” now begins a new paragraph. This separation enhances the logical transition between receptor dynamics and the local production of estradiol in adipose tissue.
Comment 3: We appreciate your guidance. The PDF you provided was essential in clarifying your request. Following your recommendation, we have integrated the two subfigures into a single, unified Figure 2. The updated version now presents a more coherent and simplified overview of the hormonal, immune, and metabolic axes contributing to lipedema progression. Inflammatory and fibrotic elements that were previously duplicated have been consolidated to avoid redundancy. Importantly, we have explicitly included progesterone resistance in the hormonal pathway, reflecting its central role in the disease’s pathophysiology.
The previously separate block listing “edema, fibrosis, pain, adipocyte hypertrophy” has been removed, as it no longer serves a clear purpose within the new layout. This reformulation aligns the figure more directly with the therapeutic targets and clinical outcomes discussed in the manuscript.
Round 4
Reviewer 1 Report
Comments and Suggestions for Authors I think the manuscript has improved a lot. Thanks for the author's efforts.